# Trends in the Use of Second-Generation Androgen Receptor Axis Inhibitors for Metastatic Hormone-Sensitive Prostate Cancer and Clinical Factors Predicting Biological Recurrence

**DOI:** 10.3390/diagnostics13091661

**Published:** 2023-05-08

**Authors:** Keita Nakane, Hiromitsu Watanabe, Taku Naiki, Kiyoshi Takahara, Takahiro Yasui, Hideaki Miyake, Ryoichi Shiroki, Takuya Koie

**Affiliations:** 1Department of Urology, Gifu University Graduate School of Medicine, Gifu 5011194, Japan; goodwin@gifu-u.ac.jp; 2Department of Urology, Hamamatsu University School of Medicine, Hamamatsu 4313192, Japan; urohiro@hama-med.ac.jp (H.W.); hmiyake@hama-med.ac.jp (H.M.); 3Department of Nephro-urology, Graduate School of Medical Sciences, Nagoya City University, Nagoya 4678601, Japan; naiki@med.nagoya-cu.ac.jp (T.N.); yasui@med.nagoya-cu.ac.jp (T.Y.); 4Department of Urology, Fujita Health University School of Medicine, Toyoake 4701192, Japan; takahara@fujita-hu.ac.jp (K.T.); rshiroki@fujita-hu.ac.jp (R.S.)

**Keywords:** metastatic hormone-sensitive prostate cancer, androgen receptor axis targeted agent, abiraterone acetate, apalutamide, enzalutamide, biochemical recurrence-free survival, neutrophil-to-lymphocyte ratio, prostate-specific antigen

## Abstract

The advent of second-generation androgen receptor axis-targeted agents (ARATs) has revolutionized the treatment of metastatic hormone-sensitive prostate cancer (mHSPC). Biochemical recurrence-free survival (BRFS) was used to compare the efficacy of each ARAT. This multicenter retrospective study included 581 patients with newly diagnosed mHSPC who received first-line hormone therapy. The characteristics of patients treated with different ARATs were compared as well as changes in the usage of each drug over time. For BRFS, the apalutamide (Apa) and enzalutamide (Enza) groups, as well as the abiraterone acetate (Abi) and Apa/Enza groups, were compared. In addition, multivariate analysis was performed to determine predictive factors for biochemical recurrence (BCR). The use of second-generation ARATs tended to increase after May 2020. No significant difference in BRFS was found between patients receiving Apa and Enza (*p* = 0.490) and those receiving Abi or Apa/Enza (*p* = 0.906). Multivariate analysis revealed that the neutrophil-to-lymphocyte ratio (NLR) ≥ 2.76 and PSA ≥ 0.550 ng/mL were independent predictors of BCR. There were no significant differences in patient characteristics or BRFS in patients with mHSPC receiving different ARATs as first-line treatment. NLR and PSA may be prognostic factors following the first-line treatment of patients with mHSPC.

## 1. Introduction

Until over a decade ago, the standard of care for metastatic, hormone-sensitive prostate cancer (mHSPC) was androgen deprivation therapy (ADT), including orchiectomy or gonadotropin-releasing hormone (GnRH) agonist/antagonist, or combined androgen blockade with GnRH plus bicalutamide (Bica) [1,2]. Although approximately 90% of patients respond to these treatments, in most cases, the disease progresses to castration-resistant prostate cancer (CRPC) within 1–3 years [3]. Furthermore, it has been reported that patients with metastatic CRPC will die of prostate cancer within 3–4 years regardless of the subsequent treatment [4]. Therefore, it became necessary to provide more aggressive first-line treatment for patients diagnosed with mHSPC. The GETUG-AFU-15 and CHAARTED studies evaluated the efficacy and tolerability of docetaxel (DOC) as an add-on to ADT in patients with mHSPC to provide a more potent potential first-line treatment [3,5]. The GETUG-AFU-15 trial showed an improvement in progression-free survival (PFS) as the primary endpoint but not in overall survival (OS) when ADT + DOC combination therapy was compared with ADT alone [3]. Similarly, the CHAARTED trial suggested that combination ADT and DOC therapy could improve OS as well as PFS in patients with mHSPC [5]. Subsequently, various randomized phase III clinical trials of second-generation androgen receptor axis-targeted inhibitors (ARATs) in patients with mHSPC, including the LATITUDE [6] and STAMPEDE studies [7] with abiraterone (Abi), the TITAN study with apalutamide (Apa) [8], and the ENZAMET [9] and ARCHES [10] studies with enzalutamide (Enza) were conducted. The results revealed that the early administration of ARATs improved oncological outcomes such as OS, PFS, radiographic progression-free survival, and treatment failure-free survival (TFFS) [6,7,8,9,10]. As such, ADT plus ARATs became the first-line treatment for mHSPC.

Although various ARATs are currently available to treat mHSPC, no randomized controlled trials directly comparing the efficacy and safety of ARATs have been undertaken. Therefore, no definitive conclusions have been obtained regarding which ARATs are most appropriate for specific patients with mHSPC or the differences in oncologic outcomes when different ARATs are used. Furthermore, it is unclear how treatment methods have changed in real-world practice since 2020 when ARATs became available for use in patients with mHSPC in Japan. In addition, research regarding factors that predict treatment efficacy is lacking. Therefore, we designed a multicenter retrospective study to evaluate patient characteristics and oncologic outcomes as well as to examine clinical factors that predict BRFS.

## 2. Materials and Methods

### 2.1. Patient Population

This study was approved by the Institutional Review Boards of the four participating institutions (authorization numbers: HM20-465, 60-21-0018, 2021-042, and 21-051). Written informed consent was not obtained from the enrolled patients because of the retrospective study design. In Japan, retrospective or observational studies are required to disclose research information, including existing materials, in accordance with the provisions of ethics committees and ethical guidelines. The summary of this trial can be reviewed by accessing the following: https://www.med.gifu-u.ac.jp/visitors/disclosure/docs/2021-042.pdf (accessed on 28 February 2023).

We retrospectively analyzed 581 patients with untreated mHSPC who were newly diagnosed and initiated with hormone therapy from February 2018 to December 2021 at Gifu University Hospital, Nagoya City University Hospital, Fujita Medical University Hospital, and Hamamatsu University Hospital, and their affiliated institutions belonging to the Tokai Urologic Oncology Research Seminar (TOURS) group. All enrolled patients underwent prostate biopsy and had pathologically confirmed prostate cancer. All patients with mHSPC underwent computed tomography (CT) of the chest, abdomen, and pelvis, as well as magnetic resonance imaging (MRI) of the pelvis, and bone scintigraphy for evaluation of pretreatment baseline characteristics. The American Joint Committee on Cancer Staging Manual was used to classify the tumor stage [11].

Pretreatment clinical information included patient age, Eastern Cooperative Oncology Group performance status (ECOG-PS) [12], initial prostate-specific antigen (PSA), biopsy Gleason score (GS), clinical T stage, lymph node involvement, metastatic site, number of metastases, hemoglobin, white blood cell count, neutrophil count, lymphocyte count, neutrophil-to-lymphocyte ratio (NLR), alkaline phosphatase (ALP), lactate dehydrogenase, albumin, and C-reactive protein (CRP) were collected as clinical data before the initiation of hormone therapy for mHSPC. PSA nadir, the date of PSA nadir, and the date of biochemical or radiological recurrence following first-line treatment were also collected.

The patients were divided into the Abi, Apa, and Enza groups based on the type of ARAT administered. The definition of biochemical or clinical disease progression was according to the Prostate Cancer Clinical Trials Working Group 3 criteria [13]. High-risk prostate cancer as defined in the LATITUDE trial was considered to be patients with mHSPC with at least two of the following: (i) ≥3 bone metastases observed by bone scintigraphy, (ii) Gleason sum ≥ 8, and (iii) visceral metastasis (LATITTUDE high-risk) [6].

### 2.2. Statistical Analysis

We investigated how the agents used for treatment changed before and after the availability of ARAT as upfront treatment for mHSPC. For BRFS, the Apa and Enza groups were compared in all enrolled patients. For LATITTUDE high-risk patients, we compared the BRFS between the Abi group and those who received Apa or Enza. Statistical analysis was conducted using EZR (Saitama Medical Center, Jichi Medical University, Saitama, Japan), which is a GUI of R (The R Foundation for Statistical Computing, Vienna, Austria). Comparisons between two or three groups were assessed using the Mann–Whitney U test or Kruskal–Wallis test for continuous variables. Changes for each category were compared using Pearson’s chi-square test or Fisher’s exact test. The cutoff value of the clinical covariate, based on the area under the receiver operating characteristic (ROC) curve, was defined as the minimum value of (1 − sensitivity)^2^ + (1 − specificity)^2^ [14]. BRFS was evaluated utilizing the Kaplan–Meier method. Differences depending on clinical variables were assessed using the log-rank test. Univariate and multivariate logistic regression analyses were performed to determine predictive factors for biochemical recurrence (BCR). All *p*-values were two-tailed, and statistical significance was set at *p* < 0.05.

## 3. Results

### 3.1. Trends in ARAT Agent Use

Table 1 presents the patient characteristics for each drug used to treat mHSPC. Among the enrolled patients, over 80% possessed a GS ≥ 8 and ≥3 bone metastases and were classified as LATITTUDE high-risk. The number of bone metastases was significantly higher in the Enza group than in the Apa group (*p* = 0.006). There was no significant difference between the Abi group and the other two groups regarding LATITTUDE high-risk.

Patients were divided into two groups based on whether or not they fulfilled the high-risk criteria of the LATITTUDE study (LATITTUDE high-risk and non-high-risk groups). Figure 1 shows the evolution of the agents used to treat patients with mHSPC in each group around the year 2020. In the LATITUDE non-high-risk group, Bica was used in 94.7% of patients before 2020. After 2020, Bica use in this group decreased to 63.1%, whereas Apa and Enza were used in 25 and 10.5% of patients, respectively. In the LATITUDE high-risk group, 54.6% of patients received Bica and 44.8% received Abi before 2020. After 2020, however, the agents used to treat mHSPC changed drastically, with 24.3%, 28.2%, 28.2%, and 19.3% of patients treated with Bica, Abi, Apa, and Enza, respectively.

The incidence of adverse events (AEs) was examined for cardiovascular disease, fatigue, hot flashes, hypokalemia, hepatic dysfunction, seizures, and rash in patients with mHSPC who received each ARAT. Although the groups did not differ with respect to the other AEs, the Apa group had a significantly higher incidence of rash than the other two groups (*p* < 0.001).

### 3.2. Comparison of Progression-Free Survival according to ARAT

Overall, the 1-year BRFS rate was 86.5% in the Apa group and 86.2% in the Enza group, but the difference was not significant (*p* = 0.490; Figure 2A). In the LATITUDE high-risk patients, the 1-year BRFS rates were 80.4 and 85.5% in the Apa and Enza groups, respectively. Similarly, there was no clinically meaningful difference between both groups (*p* = 0.913; Figure 2B).

In the LATITTUDE high-risk group after 2020, the 1-year BRFS rate was 84.0% in the Abi group and 81.8% in the Apa or Enza group with no significant difference between the two groups (*p* = 0.965, Figure 3).

### 3.3. Uni- and Multivariate Analyses for Factors Predicting BCR

Uni- and multivariate analyses were performed to determine clinical factors for predicting BCR in patients treated with ARAT for mHSPC (Table 2). On multivariate analysis, NLR and PSA at 3 months after treatment were independent predictors for BCR. Regarding NLR, the AUC was used to examine the cutoff value for predicting BCR. The AUC at an NLR of 2.76 was 0.683 (95% confidence interval (CI): 0.546–0.82), which was adopted as the cutoff value when considering the BCR in patients with mHSPC.

Using NLR and PSA at 3 months after treatment as predictors of BCR identified in a multivariate analysis, we examined BRFS using the Kaplan–Meier method in patients with mHSPC treated with ARATs (Figure 4). The patients were divided into two groups: those who had none or one of the aforementioned factors (0–1 predictor group) and those with two factors (2-factor group). The 1-year BRFS rate was 95.7% in the 0–1 predictor group and 55.5% in the 2-factor group. BCR in the 2-factor group was found to be significantly higher after ARAT treatment than in the 0–1 predictor group (*p* < 0.001).

## 4. Discussion

It has been recently reported that the time taken for mHSPC to progress to mCRPC is an important prognostic factor [4]. Therefore, the current choice of primary therapy for mHSPC would be of paramount importance [15,16]. Indeed, treatment for mHSPC appears to be undergoing a revolution with the advent of second-generation ARAT agents. Two large-phase III randomized controlled trials, namely the LATITTUDE and STAMPEDE trials, were conducted to investigate the efficacy of Abi for mHSPC [6,7]. The LATITTUDE trial examined OS and radiographic PFS (rPFS) as primary endpoints in patients who received ADT plus Abi and in those who received ADT plus placebo [6]. The OS and rPFS were significantly longer in patients receiving ADT + Abi compared with those receiving ADT + placebo (*p* < 0.001 and *p* < 0.001, respectively) [6]. It should be carefully noted, however, that the patients who received Abi were in the LATITTUDE high-risk group [6]. The STAMPEDE trial compared the efficacy of Abi with OS and TFFS as primary endpoints in patients with mHSPC [7]. Death from PCa occurred in 19.2% of patients treated with ADT plus Abi and 27.4% of those treated with ADT alone. Cancer death was significantly lower in patients treated with ADT plus Abi compared to those treated with ADT alone (hazard ratio (HR): 0.63; 95% CI: 0.52–0.76; *p* < 0.001) [7]. Similarly, TFFS was significantly prolonged in the Abi group compared with that in the ADT alone group (HR: 0.29; 95% CI: 0.25–0.34; *p* < 0.001) [7]. In the TITAN study with OS and rPFS as primary endpoints, both OS and rPFS were significantly prolonged in patients treated with ADT plus Apa compared with those treated with ADT plus placebo (*p* = 0.005 and *p* < 0.001, respectively), although no statistical difference was observed regarding AEs between the two groups [8]. The ENZAMET and ARCHES trials evaluated the efficacy of Enza in patients with mHSPC [9,10]. Although the ENZAMET and ARCHES phase III trials were conducted with PFS and OS as primary endpoints in ENZAMET, and rPFS as the primary endpoint in the ARCHES trial, both trials demonstrated a benefit with ADT plus Enza compared with ADT alone [9,10]. In our previous study, we showed that Abi in patients with mHSPC could potentially prolong PFS and OS [15]. We also showed that ADT plus Abi significantly prolonged PFS and time to second progression compared to ADT plus Bica [16]. Therefore, the importance of ARATs as therapeutic agents for mHSPC is expected to garner greater attention in the near future.

It is unclear how many patients are treated with ARATs as therapeutic agents for mHSPC in daily clinical practice. In 2020, the Adelphi Prostate Cancer Disease Specific Programme^TM^ collected information on patients with mHSPC in the US, Europe, and Japan to determine which agents are used as first-line treatment [17]. ADT was used as first-line therapy in 47.2% of patients with mHSPC in the overall population and 78.4% in the Japanese population [17]. On the other hand, ARATs were used in 28.9% of the overall population and 19.2% of the Japanese population, suggesting that ADT is preferred as first-line treatment for mHSPC, especially in Japan [17]. We investigated the transition of treatment agents for mHSPC before and after 2020. In the LATITTUDE non-high-risk group, the use of Bica decreased significantly from 94.7% to 63.1%. After 2020, Apa and Enza were used in 25% and 10.5% of cases, respectively. For the LATITUDE high-risk group, Bica was administered to 54.6% of patients with mHSPC and Abi was administered to 44.8% before 2020. After 2020, Bica was used in 24.3% of cases, Abi was used in 28.2%, Apa was used in 28.2% and Enza was used in 19.3%. Enza was used in 19.2% of those with mHSPC. These results indicate that although ARATs have been used in many patients with mHSPC in Japan since 2020, some patients are still administered Bica, and ARATs are used in a variety of therapeutic modalities.

To clarify the selection criteria for ARATs in practice, clinical factors of patients with mHSPC treated with each ARAT were investigated. In the LATITTUDE high-risk group, there were no significant differences in patient background among the three groups. A comparison of patients with mHSPC in the LATITTUDE non-high-risk group who received Apa and Enza revealed that significantly more patients with ≥3 bone metastases received Enza compared to those who received Apa. PFS comparisons of these groups showed no significant differences between patients with mHSPC who received Apa and Enza. The TITAN [8] and ENZAMET trials [9] using Apa and Enza all adopted the definition of high-volume PCa used in the CHAARTED trial: namely, the presence of visceral metastases or at least four bone lesions with one beyond the vertebral bodies and pelvis, metastases or at least four bone lesions with one beyond the vertebral bodies and pelvis [18]. In a subgroup analysis of the final report of the TITAN trial, the HR for risk of death with Apa was 0.70 (95% CI: 0.56–0.88) in the high-volume group and 0.52 (95% CI: 0.35–0.71) in the low-volume group compared to those of ADT alone [19]. Regarding the number of bone metastases, the HR was 0.69 (95% CI: 0.52–0.93) for ≤10 metastases and 0.54 (95% CI: 0.42–0.71) for ≥11 metastases, both results indicating the usefulness of Apa [19]. A discrepancy of HR between tumor volume and the number of bone metastases was observed, suggesting that Apa may be more effective in the treatment of mHSPC, especially for those with a high number of bone metastases [19]. Similarly, in the ENZAMET study, the subgroup analysis for OS showed an HR of 0.43 (95% CI: 0.26–0.72) for the low-volume group and 0.80 (0.59–1.07) for the high-volume group, and the subgroup analysis for PFS showed an HR of 0.30 (0.22–0.43) for the low-volume group and 0.45 (0.36–0.57) for the high-volume group [9]. These results suggest that Enza may be a candidate for treatment in patients with mHSPC who had low tumor volume when OS is the endpoint, although tumor volume may not be a consideration for PFS [9]. Therefore, the choice of ARATs to be administered for mHSPC may need to be based on the characteristics of each.

Since the tumor microenvironment, particularly the inflammatory response, may play an important role in cancer development and progression, it has been suggested that measuring clinical factors involved in inflammation, such as CRP, albumin, and white blood cells, may predict systemic cancer progression and metastasis [20]. Among them, NLR is widely known to correlate with the prognosis of many urologic cancers, including mHSPC and CRPC [21,22,23,24,25,26]. According to a subgroup analysis of the COU302 trial, which investigated the effect of Abi in patients with mCRPC, oncological outcomes were examined using NLR 2.5 as the cutoff value [24]. Although there was no significant difference in OS between the high and low NLR groups, biochemical PFS was significantly worse in the high NLR group than in the low NLR group (*p* = 0.004) [24]. In addition, the Abi group with NLR ≥ 2 from baseline showed a statistically significant improvement in OS (*p* = 0.024) and rPFS (*p* = 0.002) compared with the placebo group, although there was no change in biochemical PFS [24]. In contrast, there were no significant differences in OS, rPFS, or biochemical PFS in patients with NLR ≥ 2 from baseline with or without Abi [24]. In a Canadian study of 3556 patients with PCa and newly detected metastases, divided into five groups according to NLR values, patients with a higher NLR had a significantly increased relative risk of PCa death compared to those with a lower NLR (HR, 1.55; 95% CI, 1.27–1.90) [25]. In the study that examined the usefulness of NLR as a predictor of PCa diagnosis and disease progression, the probability of PCa metastasis was significantly increased in patients with high NLR, and the risk of metastasis was reported to be 3.2 times higher in patients with NLR > 2.5 [26]. Based on our findings, we suggest that NLR may be a useful prognostic factor for patients with mHSPC treated with ARATs.

Alterations in PSA, particularly the PSA nadir, is recognized as a useful indicator of treatment response or a predictor of prognosis in patients with PCa receiving systemic therapy [27,28,29]. The post hoc analysis of the LATITTUDE trial revealed that time to PSA progression was significantly longer in patients receiving ADT plus Abi than in those receiving ADT plus placebo (HR, 0.3; *p* < 0.001) and was associated with rPFS and OS [27]. The risk of PCa death was also reduced in patients with PSA declines of 50% and more than 90% compared with baseline after the administration of Abi [27]. Achieving PSA ≤ 0.1 ng/mL and the time to PSA nadir were strongly associated with rPFS and OS [27]. Tomioka et al. [28] reported a study of 286 patients with metastatic PCa who received ADT as first-line treatment with the proportion of patients with CRPC and OS as primary endpoints. In a multivariate analysis on CRPC, GS, nadir PSA, and the time from the initiation of ADT to achieving nadir PSA were significant independent factors [28]. Regarding OS, nadir PSA and the time from the initiation of ADT to achieving the nadir PSA were identified as prognostic factors [28]. A study of 248 patients with PCa with bone metastases reported that high ALP, short time to PSA nadir, and pain were associated with significantly increased risk of progression to CRPC, and that ALP, ECOG-PS, and PSA nadir were independent predictors of cancer-specific survival [29]. Therefore, since PSA nadir seems to be one of the prognostic factors for patients with mHSPC, it may be necessary to carefully monitor how PSA declines after the initiation of hormone therapy.

Our findings revealed that the results for PFS were similar for all risk categories regardless of which ARATs were administered in patients with mHSPC (Figure 2 and Figure 3). However, based on the results obtained in the multivariate analysis, stratification by NLR and PSA levels at 3 months after the administration of ARAT showed that PFS was significantly worse in patients with both compared to the counterpart group (*p* < 0.001; Figure 4). Therefore, these two markers may be used to predict the therapeutic effect of ARATs in mHSPC or may indicate a change from primary to secondary therapy.

Several limitations exist in this study. First, there is a potential bias in the patient selection criteria because of the multicenter, retrospective nature of the study. Second, owing to the relatively small number of patients enrolled and the short median follow-up period, caution must be exercised in interpreting the findings of this study. Third, the selected agents for the treatment of mHSPC depend on the patient preferences and the discretion of the attending physician, and there are no defined criteria. Fourth, biopsy specimens were not re-evaluated by a central pathologist; hence, there is a possibility that variations in diagnosis could exist among institutions, especially regarding the GS. Fifth, since combination therapy with docetaxel, darolutamide, and ADT has been approved for patients with mCSPC in Japan from 2023, we did not collect cases regarding this combination therapy in the present study. Sixth, this study did not evaluate the superiority of any particular drug with respect to therapeutic efficacy. Finally, the prognostic factors identified in this study result from retrospective data collection. Prospective studies with large cohorts and long-term follow-up are required to confirm their true utility.

## 5. Conclusions

This study confirmed a trend toward the use of agents in the treatment of mHSPC. There was no significant difference in PFS regardless of which ARAT drug was used for mHSPC. Multivariate analysis identified predictive factors for PFS. A large prospective study should be conducted to validate the results of this study.

## Figures and Tables

**Figure 1 diagnostics-13-01661-f001:**
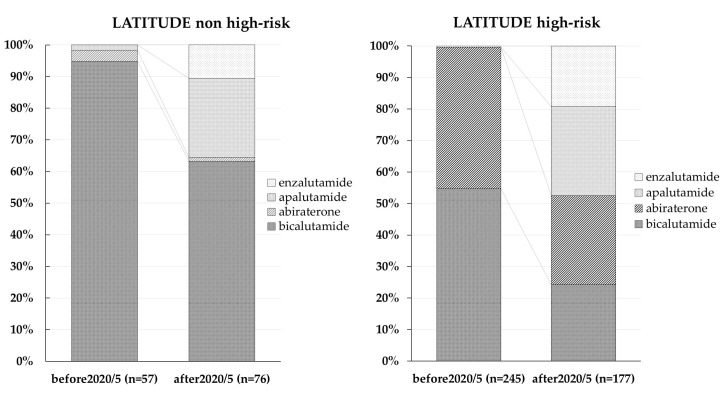
Evolution of agents used to treat patients who did not fulfill the high-risk criteria of the LATITTUDE study (LATITTUDE non-high-risk group) and those who did (LATITTUDE high-risk group) around the year 2020.

**Figure 2 diagnostics-13-01661-f002:**
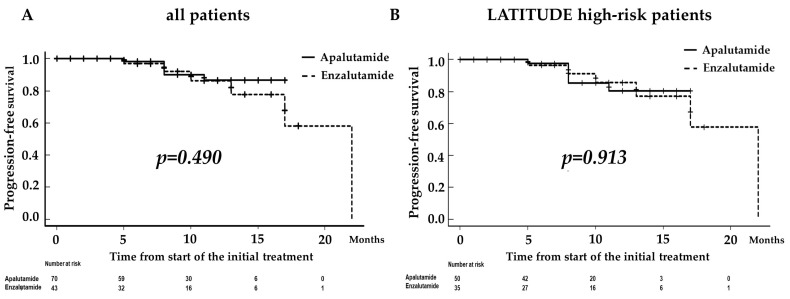
Kaplan–Meier estimates of biochemical recurrence-free survival (BRFS) for all enrolled patients (**A**) and those who met the criteria for high-risk prostate cancer (PCa) in the LATITTUDE trial (**B**). The 1-year BRFS rate was 86.5% in patients receiving apalutamide (the Apa group) and 86.2% in those treated with enzalutamide (the Enza group), but the difference was not significant (*p* = 0.490; (**A**)). In patients with high-risk PCa according to the LATITUDE study, the 1-year BRFS rates were 80.4 and 85.5% in patients receiving Apa and Enza, respectively. Similarly, there was no significant clinically meaningful difference between the groups (*p* = 0.913; (**B**)).

**Figure 3 diagnostics-13-01661-f003:**
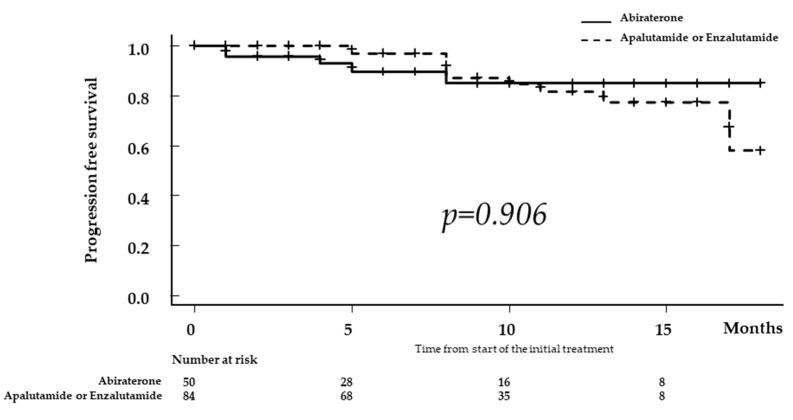
Kaplan–Meier estimates of biochemical recurrence-free survival (BRFS) in patients who met the high-risk prostate cancer criteria according to the LATITTUDE study. The BRFS rate was 84.0% in patients receiving abiraterone (the Abi group) and 81.8% in those treated with apalutamide or enzalutamide (the Apa or Enza group), with no significant difference between the two groups (*p* = 0.965, Figure 3).

**Figure 4 diagnostics-13-01661-f004:**
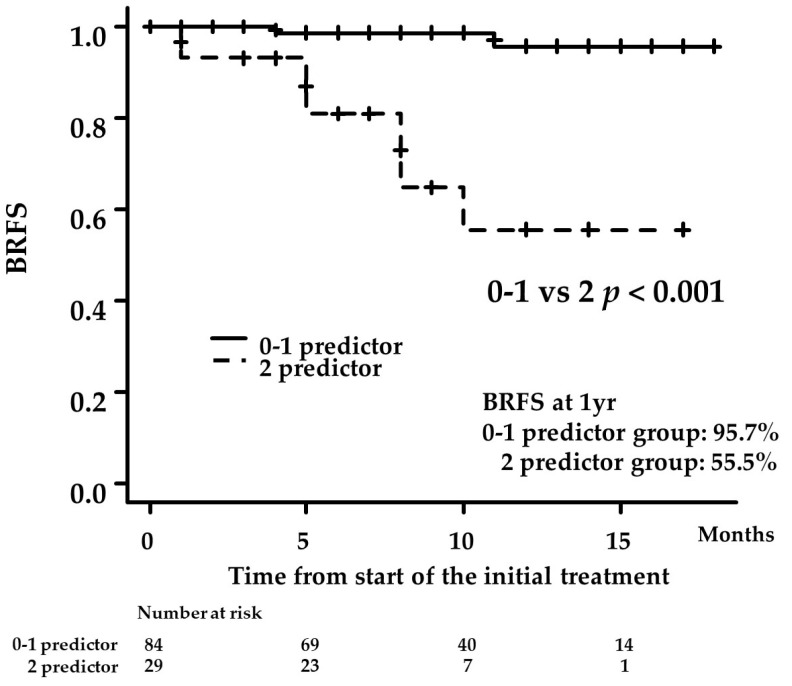
Kaplan–Meier estimates of biochemical recurrence-free survival (BRFS) according to the patients with neutrophil-to-lymphocyte ratio and prostate-specific antigen as factors) 3 months after the administration of androgen receptor axis inhibitors (2-factor group) or those with none or one factor (0–1-factor group). The 1-year BRFS rate was 95.7% in the 0–1 predictor group and 55.5% in the 2-factor group (*p* < 0.001).

**Table 1 diagnostics-13-01661-t001:** Patient characteristics for each agent used to treat patients with mHSPC.

Variables	All	Abiraterone	Apalutamide	Enzalutamide	*p*-Value
Number of patients	276	163	70	43	
Age (years, median, IQR)	73.0 (68.0–79.0)	73.0 (68.0–79.0)	73.5 (69.0–78.7)	73.0 (65.5–76.0)	0.575
ECOG-PS (number, %)					0.792
0	198 (71.7)	117 (71.8)	51 (72.9)	30 (69.8)	
1	46 (16.7)	24 (14.7)	13 (18.6)	9 (20.9)	
2	27 (9.8)	19 (11.7)	5 (7.1)	3 (7.0)	
3	3 (1.1)	2 (1.2)	0 (0.0)	1 (2.3)	
4	2 (0.7)	1 (0.6)	1 (1.4)	0 (0.0)	
Initial PSA (ng/mL, median, IQR)	286.5 (68.7–1163.2)	409.0 (78.8–1656.0)	185.9 (57.5–598.3)	380.4 (92.5–1011.9)	0.05
Gleason score (number, %)					0.002
≤6	0 (0.0)	0 (0.0)	0 (0.0)	0 (0.0)	
7	12 (4.3)	1 (0.6)	8 (11.6)	3 (7.0)	
≥8	262 (95.0)	161 (99.4)	61 (88.4)	40 (93.0)	
Clinical T stage (number, %)					0.927
1c	0 (0.0)	0 (0.0)	0 (0.0)	0 (0.0)	
2	42 (15.2)	24 (14.7)	11 (15.7)	7 (16.3)	
3	134 (48.6)	81 (49.6)	35 (50.0)	18 (41.9)	
4	92 (33.3)	55 (33.7)	21 (30.0)	16 (37.2)	
x	8 (2.8)	3 (1.8)	3 (4.3)	2 (4.7)	
Lymph node involvement (number, %)					0.461
Negative	111 (40.2)	62 (38.0)	30 (42.9)	19 (44.2)	
positive	165 (59.8)	101 (62.0)	40 (57.1)	24 (55.8)	
Number of bone metastasis (number, %)					<0.001
1	18 (6.5)	5 (3.1)	11 (16.2)	2 (5.0)	
2	11 (3.9)	2 (1.2)	7 (10.3)	2 (5.0)	
≥3	225 (81.5)	144 (88.9)	45 (66.2)	36 (90.0)	
Visceral metastasis (number, %)	93 (33.7)	59 (36.2)	23 (32.9)	11 (25.6)	0.418
Patients who met the criteria for high-risk PCa in the LATITTUDE trial (number, %)	245 (89.2)	160 (98.2)	50 (71.4)	35 (81.4)	<0.001
WBC (count/μL, median, IQR)	6680 (5500–7860)	6600 (5470–7810)	6690 (5500–8280)	6800 (5800–7800)	0.668
Neutrophil (count/μL, median, IQR)	4100 (3235–5311)	4100 (3200–5140)	4200 (3235–5444)	4000 (3373–5587)	0.816
Lymphocyte (count/μL, median, IQR)	1500 (1191–1892)	1500 (1200–1803)	1400 (1096–1911)	1440 (1247–2229)	0.584
NLR	2.74 (2.00– 3.90)	2.73 (2.00–3.82)	2.77 (1.76–4.49)	2.54 (1.92–3.38)	0.860
ALP (U/L, median, IQR)	275 (174– 605)	401 (235–923)	213 (101–310)	158 (102–299)	<0.001
LDH (U/L, median, IQR)	204 (178– 243)	205 (178–248)	198 (169–229)	206 (182–242)	0.397
Albumin (g/dL, median, IQR)	4.0 (3.6– 4.3)	3.9 (3.5–4.2)	4.1 (3.7–4.3)	4.0 (3.8–4.3)	0.099
CRP (mg/dL, median, IQR)	0.23 (0.06–1.06)	0.26 (0.08–1.27)	0.20 (0.04–1.15)	0.19 (0.06–0.81)	0.487
Follow-up period(months, median, IQR)	12.0 (6.0–22.5)	19.0 (8.5–27.0)	9.0 (7.0–12.0)	7.0 (4.5–12.5)	<0.001

IQR, interquartile range; ECOG-PS, Eastern Cooperative Oncology Group performance status; PSA, prostate-specific antigen; PCa, prostate cancer; WBC, white blood cell; NLR, neutrophil-to-lymphocyte ratio; ALP, alkaline phosphatase; LDH, lactate dehydrogenase; CRP, C-reactive protein.

**Table 2 diagnostics-13-01661-t002:** Uni- and multivariate analyses.

	Univariate Analysis			Multivariate Analysis		
	HR	95% CI	*p*-Value	HR	95% CI	*p*-Value
Age (≥73 vs. <73 years)	3.040	0.847–10.900	0.087	-	-	-
ECOG-PS (≥1 vs. 0)	2.140	0.741–6.176	0.159	-	-	-
Albumin (≥3.8 vs. <3.8 g/dL)	0.351	0.120–1.077	0.068	-	-	-
NLR (≥2.76 vs. <2.76)	11.690	1.495–91.480	0.019	9.700	1.2010–78.360	0.033
Hemoglobin (≥13.1 vs. <13.1 g/dL)	0.450	0.150–1.344	0.153	-	-	-
Gleason score 5 (yes vs. no)	1.473	0.461–4.703	0.513	-	-	-
Initial PSA (≥413 vs. <413 ng/mL)	0.745	0.249–2.226	0.599	-	-	-
Time to PSA nadir (≥ 6 vs. <6 months)	0.161	0.042–0.602	0.007	-	-	-
PSA level 3 months after the start of first-line treatment (≥0.55 vs. <0.55 ng/mL)	7.191	1.608–32.170	<0.001	10.840	1.3620–86.360	0.024

ECOG-PS: Eastern Cooperative Oncology Group performance status, NLR: neutrophil-to-lymphocyte ratio, PSA: prostate-specific antigen.

## Data Availability

The data presented in this study are available on request from the corresponding author. The data are not publicly available due to privacy and ethical reasons.

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
