# Peer review of "Trends in the Use of Second-Generation Androgen Receptor Axis Inhibitors for Metastatic Hormone-Sensitive Prostate Cancer and Clinical Factors Predicting Biological Recurrence"

_diagnostics, 2023, doi:10.3390/diagnostics13091661_

Round 1

Reviewer 1 Report

Prostate cancer is one of the most common causes of cancer death in men. In their manuscript “Trends in the use of Second-Generation Androgen Receptor Axis-Targeted Agents for Metastatic Hormone-Sensitive Prostate Cancer and Clinical Factors Predicting Biological Recurence”, Nakane et al. analyze effectiveness of second-generation androgen receptor axis-targeted agents (ARATs) in treatment of hormone-sensitive prostatic cancer (mHSPC). In their retrospective multicenter studies, they analyzed effects of use of various ARATs as well as transition of treatment agents in 581 patients. They propose neutrofhil-to-lymphocyte ratio and prostate specific antigen as an independent treatment prediction factor for biochemical recurrence. The manuscript is well structured and well written, can be accepted in present form.

Author Response

Response:

The authors appreciate the reviewer’s comments.

Reviewer 2 Report

This is an interesting retrospective comparative study on the use of different ARTAs for mHSPC. Some points that need to be addressed are listed below:

- the authors have conducted this study prior to the era of triplet therapy particularly for high volume pts. How many pts qualified for triplet therapy within their study?

- how did the cut off for NLR come up?

- were there any differences in toxicity profiles between different ARTAs?

- the authors need to highlight that because of the retrospective natire of the study no definitive conclusions can be drawn regarding preferences of one ARTA over another.

Author Response

Response to Reviewer 2

The authors appreciate the Academic Editor’s comments. The authors’ point-by-point responses to the comments are given below.

This is an interesting retrospective comparative study on the use of different ARTAs for mHSPC. Some points that need to be addressed are listed below:

  1. The authors have conducted this study prior to the era of triplet therapy particularly for high volume pts. How many pts qualified for triplet therapy within their study?

Response:

The authors have added the following sentence on line 331:

Fifth, since combination therapy with docetaxel, darolutamide, and ADT has been approved for patients with mCSPC in Japan from 2023, we did not collect cases regarding this combination therapy in the present study.

  1. How did the cut off for NLR come up?

Response:

The authors have added the following sentence on line 178:

Regarding NLR, the AUC was used to examine the cutoff value for predicting BCR. The AUC at an NLR of 2.76 was 0.683 (95% confidence interval [CI]: 0.546-0.82), which was adopted as the cutoff value when considering the BCR in patients with mHSPC.

  1. Were there any differences in toxicity profiles between different ARTAs?

Response:

The authors have added the following sentences on line 144:

The incidence of adverse events (AEs) was examined for cardiovascular disease, fatigue, hot flashes, hypokalemia, hepatic dysfunction, seizures, and rash in patients with mHSPC who received each ARAT. Although the groups did not differ with respect to the other AEs, the Apa group had a significantly higher incidence of rash than the other two groups (p < 0.001).

  1. The authors need to highlight that because of the retrospective nature of the study no definitive conclusions can be drawn regarding preferences of one ARTA over another.

Response:

The authors have added the following sentences on line 333:

Sixth, this study did not evaluate the superiority of any particular ARATs with respect to therapeutic efficacy in patients with mHSPC.

Round 2

Reviewer 2 Report

No additional comments